# Nano-Sized Selenium Maintains Performance and Improves Health Status and Antioxidant Potential While Not Compromising Ultrastructure of Breast Muscle and Liver in Chickens

**DOI:** 10.3390/antiox12040905

**Published:** 2023-04-10

**Authors:** Damian Bień, Monika Michalczuk, Małgorzata Łysek-Gładysińska, Artur Jóźwik, Anna Wieczorek, Arkadiusz Matuszewski, Misza Kinsner, Paweł Konieczka

**Affiliations:** 1Department of Animal Breeding, Institute of Animal Sciences, Warsaw University of Life Sciences WULS-SGGW, 02-786 Warszawa, Poland; 2Division of Medical Biology, Institute of Biology, University of Jan Kochanowski, Uniwersytecka 7, 25-406 Kielce, Poland; 3Institute of Genetics and Animal Breeding PAS, Jastrzębiec, Postępu 36A, 05-552 Magdalenka, Poland; 4Department of Animal Environment Biology, Institute of Animal Sciences, Warsaw University of Life Sciences WULS-SGGW, 02-786 Warszawa, Poland; 5Department of Animal Nutrition, The Kielanowski Institute of Animal Physiology and Nutrition, Polish Academy of Sciences, Instytucka 3, 05-110 Jabłonna, Poland; 6Department of Poultry Science and Apiculture, University of Warmia and Mazury in Olsztyn, 10-719 Olsztyn, Poland

**Keywords:** chickens, selenium, nanoparticles, quality of meat, liver, ultrastructure, antioxidant potential, health status

## Abstract

The poultry industry is looking for the most effective sources of selenium (Se) for commercial use. Over the past five years, nano-Se has attracted a great deal of attention in terms of its production, characterisation and possible application in poultry production. The objective of this study was to evaluate the effects of dietary levels of inorganic and organic Se, selenised yeast and nano forms of selenium on breast meat quality, liver and blood markers of antioxidants, the ultrastructure of tissue and the health status of chickens. A total of 300 one-day-old chicks Ross 308 were divided into 4 experimental groups, in 5 replications, with 15 birds per replication. Birds were fed the following treatments: a standard commercial diet containing inorganic Se in the form of inorganic Se at the level of 0.3 mg/kg diet and an experimental diet with an increased level of Se (0.5 mg/kg diet). The use of other forms of Se (nano-Se) versus sodium selenate significantly influences (*p* ≤ 0.05) a higher collagen content and does not impair physico-chemical properties in the breast muscle or the growth performance of the chickens. In addition, the use of other forms of selenium at an increased dose versus sodium selenate affected (*p* ≤ 0.01) the elongation of sarcomeres in the pectoral muscle while reducing (*p* ≤ 0.01) mitochondrial damage in hepatocytes and improving (*p* ≤ 0.05) oxidative indices. The use of nano-Se at a dose of 0.5 mg/kg feed has high bioavailability and low toxicity without negatively affecting the growth performance and while improving breast muscle quality parameters and the health status of the chickens.

## 1. Introduction

Selenium is an essential mineral element with important biological functions for the whole body, through incorporation into at least 30 selenoproteins [1,2,3,4]. The effect of selenium on the health of birds depends on the form in which it is supplied to the diet. Therefore, the poultry industry is currently searching for the most effective sources of Se for commercial use [5]. In animal nutrition, Se is most commonly used in two forms: inorganic, as sodium selenite (Na_2_SeO_3_) or sodium selenate (Na_2_SeO_4_), and organic, as selenomethionine (SeMet) or selenocysteine (SeCys) [6,7,8,9]. Se is an essential bio-element that, together with other micronutrients such as zinc and iodine, plays a key role in the proper functioning, development and growth of various organisms [10]. The effect of selenium on the health of birds depends on the form in which it is supplied to the diet. Inorganic Se is passively absorbed from the intestine through a simple diffusion process and competes with many mineral elements for absorption pathways, and organic Se is actively absorbed through amino acid transport mechanism and has a higher bioavailability than the inorganic form [11]. The combination of SeMet + sodium selenite is more efficient than their individual treatments for Se deposition in egg and chicken embryo tissues. Although many studies have shown that organic Se is easier to absorb than inorganic Se, the reduction of competitive absorption leads to higher absorption efficiency and production performance of the combination of the two Se sources [12]. A thorough understanding of this mechanism requires further research and analysis. Se alleviates oxidative stress and peroxidative damage to unsaturated fatty acids and influences the efficiency of fatty acid biosynthesis in animal tissues [13,14]. Dietary selenium deficiency can adversely affect the fatty acid profile and the conversion of linolenic acid (ALA) to eicosapentaenoic acid (EPA) and docosahexaenoic acid (DHA), resulting in an unfavourable n − 6/n − 3 ratio in tissue lipids [15,16,17,18,19]. Se supplementation has been found to stimulate the immune response, improve growth and reproductive performance and increase disease resistance. Se deficiency, on the other hand, increases the risk of myopathy, muscular dystrophy, causes immune declines and reduces poultry performance [20,21]. European Union regulations (list of authorised feed additives published in application of Article 9t (b) of Council Directive 70/524/EEC concerning additives in feeding stuffs (2004/C 50/01)) specify that the maximum level of Se in chicken feed must not exceed 0.5 mg/kg to ensure safe use of the feed. The requirement for Se ranges from 0.1 to 0.15 mg/kg in various poultry diets. In contrast, other authors [22,23] recommend increasing this dose to 0.3–0.75 mg/kg feed.

One of the most rapidly developing fields of science today is nanobiotechnology. The last 10 years have seen a breakthrough in research into the use of nanoparticles in poultry husbandry. Current research is mainly focused on the use of elements in nanometric form as replacements for their traditional sources [24]. Over the past five years, nano-Se has attracted a great deal of attention in terms of its production, characterisation and possible application in poultry production and other livestock species and medical science [25,26,27,28,29,30,31]. Nano-Se supplementation has variable effects compared to classical Se sources on the production performance of chickens, probably due to the variable amount of Se in the feed. However, nano-Se has significant effects on Se retention in tissues, GPx activity, or modification of the fatty acid profile in poultry, with less toxic effects. The authors of this study [32] found that the addition of nano-Se at doses up to 30 mg/kg had a significant effect on ma-weight gain and lower FCR. Comparing nanoparticles with Na_2_SeO_3_ in broiler chickens, it was found that the addition of nanoparticles in amounts of 0.2 and 0.5 mg/kg and the addition of sodium selenite (0.2 mg/kg) relative to the control group improved body weight gain and FCR, but the form of Se used had no significant effect on improving the aforementioned parameters. In a similar study, where the same Se sources were compared at four doses (0.15, 0.30, 0.60 and 1.2 mg/kg) in broiler chickens, Hu et al. [33] found that chickens had better weight gain using a dose of up to 30 mg/kg, which, however, deteriorated in later stages of rearing. With nano-Se, average daily gains and FCR improved already at a dose of 0.15 mg/kg with no change over time as with the application of Na_2_SeO_4_. The application of nano-Se at 0.5 mg/kg of feed to broiler chickens has a significant effect on the higher PUFA (polyunsaturated fatty acid) content and the protection of lipids against the action of reactive oxygen species, with its high bioavailability and low toxicity for the chicken organism. The use of higher doses of Se in feed (than the recommended 0.3 mg/kg) is a response to the increasingly rapid growth rate of the chicken body. This will provide consumers with a high-quality product rich in good-quality fats [34]. Nano-Se at a dose higher than 2 mg/kg appears to be toxic to the organisms of birds, as when studying the effects of nano-Se at a dose of 0.30 mg/kg versus 2 mg/kg, deterioration of glutathione peroxidase (GPx) activity, concentration of immunoglobulin M (the main antibodies produced during first contact of the organism with an antigen), glutathione, malondialdehyde and thus deterioration of the free radical scavenging capacity in the serum, liver, and muscle of birds was found for a dose of 2 mg/kg. Studies conducted to date [22,33] suggest that nano-Se has a higher absorption rate and antioxidant capacity at a wide range between nutritional and toxic dose compared to sodium selenite. This allows for the conclusion that the toxicity of nano-Se is even lower than even selenomethionine. 

Given the steadily increasing growth performance of broiler chickens, it seems appropriate to determine the effect of increased dietary supplementation with Se, especially Se in the form of nanoparticles. However, there is an urgent need to address issues related to the absorption, assimilation and metabolism of nano-Se in animals/poultry before it can find its way into animal/poultry production as a feed supplement [35]. The aim of this study was to compare the effects of different forms of Se, including nano-Se, on the health status, antioxidant potential and ultrastructural changes of breast muscle and liver in chickens.

## 2. Materials and Methods

### 2.1. Animals and Diets

The experiment was carried out with 300 Ross 308 chicken broilers randomly allocated to 4 experimental groups, in 5 replications, 15 birds per replication. Chickens were reared under standard conditions for 6 weeks (from hatching day until 42 days of life). They had free access to water and were kept under a controlled light cycle (according to the Aviagen Ross 308: Broiler Performance Objectives. Aviagen Inc., Huntsville, AL, USA. 2019, 1–15). For the first 10 days, all birds were fed the same starter diet balanced to meet their nutritional demands. On day 11 of life, birds started to receive respective diets. Chicken diets are presented in our previous manuscript [34]. Experimental groups differed in terms of Se form implemented in the diet, for example:

CON (control group)—diet meeting nutritional demands of Ross 308 broilers with the basic (recommended) dose of inorganic Se (SS, 0.3 mg/kg feed), 

T1 (SS)—diet with upper dose of inorganic Se (sodium selenate, 0.5 mg/kg feed),

T2 (SY)—diet with upper dose of Se in the organic form (selenised yeast, commercial preparation) (0.5 mg/kg feed),

T3 (nano-Se)—diet with upper dose of Se in the form of nanoparticles (commercial preparation) (0.5 mg/kg feed).

### 2.2. Selenium Forms Used in the Diets

In the CON treatment, a sodium selenate form was used, which was provided in the diet with a vitamin–mineral premix. This form of Se is commonly used in the formulation of diets for broiler chicken [36]. In the treatment of T2, a premix-provided Se was substituted by the Se-enriched yeast provided in the form of commercial product (SelPlex 1000, Alltech, Nicholasville, KY, USA) according to manufacturer declaration. In the treatment T3, Se was provided with the premix in the form of nanoparticles. Briefly, nano-Se in the form of nano-powder was obtained by chemical synthesis. According to manufacturer declaration (American Elements, Los Angeles, CA, USA), average particle size was 10–45 nm, specific surface area was approximately 30–50 m^2^/g, and purity was 99.9%.

### 2.3. Sampling Procedures

Forty cockerels were chosen (10 birds from each treatment; 2 birds for each replicate) for slaughter at the age of 42 days of life that had a body weight similar to the group mean. The pectoral muscle, liver, and blood samples were taken for analysis: chemical composition, physico-chemical properties, selenium content determination, indicators of health status and antioxidant potential and analysis of ultrastructure.

### 2.4. Assessment of Slaughter Efficiency, Chemical Composition, and Physico-Chemical Properties

Upon completion of the feeding experiment, 10 cockerels (n = 10) per experimental group were randomly selected and weighed before slaughter. After slaughter and cooling of the carcasses, the slaughter efficiency of the chickens was assessed based on a previous report by Michalczuk et al. [37], determining the percentages of the pectoral muscle, legs, and giblets. The basic chemical composition was determined for the collected samples of pectoral muscles: dry weight, crude fat, crude protein, and ash. The determinations were made using the NIR method [37]. The pH value, drip loss, WHC and colour parameter (L*, a*, b*) were analysed according to the method by Michalczuk et al. [38] and ΔE parameter by the method described by Bendowski et al. [39]. 

### 2.5. Selenium Content Determination

Determination of Se content in breast muscle and liver (n = 10) was performed according to the PB-28/LF method in an accredited laboratory (PCA Accreditation Certificate No. AB 1095, Issue No. 19, dated 1 January 2022). A full description of the method can be found in our previous manuscript [34]. 

### 2.6. Indicators of Health Status and Antioxidant Potential

On the day of slaughter, ten birds (n = 10) per experimental group were randomly chosen to collect blood post-mortem in the amount of 1.5 mL per bird. Laboratory analysis aimed to also determine the activity of selected enzymes and antioxidant compounds by analysing blood samples and 5 g fragments of breast muscle and liver tissue.

In order to determine the effect of the different sources and levels of Se on the health of the chickens, the activity of the following hydrolytic enzymes from the blood, liver and pectoral muscles of the chickens was determined: alanine aminopeptidase (AlaAP, EC 3.4.11.2), leucine aminopeptidase (LeuAP, EC 3.4.11.1), and arginine aminopeptidase (ArgAP, EC 3.4.11.6), which are all responsible for limiting harmful metabolism and accelerating protein circulation in the body, which translates into better weight gain. The following compounds were also determined: the activity of acid phosphatase (AcP, EC 3.1.3.2), beta-glucuronidase (BGRD, EC 3.2.1.31), beta-galactosidase (BGAL, EC 3.2.1.23), beta-glucosidase (BGLU, EC 3.2.1.21), alpha-glucosidase (aGlu, EC 3.2.1.20), mannosidase (MAN, EC 3.2.1.25), and N-acetyl-BD-hexosaminidase (HEX, EC 3.2.1.52). These compounds are responsible for breaking down complex sugars into simple sugars and for the removal of harmful metabolites formed inside the cell [39].

The activity of aminopeptidases was measured as Fast Blue BB salt derivatives at 540 nm by the method of [40], whereas the activity of AcP, BGRD, BGAL, BGLU, aGLU, aMAN, and HEX was measured as 4-nitrophenyl derivatives at 420 nm according to [41]. All determinations were performed with use of Varian Cary 50 Bio UV–VIS spectrophotometer (Santa Clara, CA, USA). The enzyme activity was expressed in nmol/mg protein/h.

The effect of Se type and dose on the oxidative status in analysed tissues of broiler chickens was tested using the following determinations: vitamin C, glutathione (GSH), and 2,2-diphenyl-1-picrylhydrazyl (DPPH). The concentration of vitamin C in the collected tissues was determined using a LambdaBio-20 spectrophotometer (Perkin Elmer, Waltham, MA, USA), whereas the level of GSH was determined by means of the OxisReasearch™ Bioxytech^®^ GSH/GSSG—412™ test (Foster City, CA, USA) according to the methods described by [42]. Measuring the radical scavenging activity was carried out using a routine test procedure employing the synthetic radical DPPH [42,43].

### 2.7. Analysis of Ultrastructure

Immediately after chicken slaughter, small fragments of the pectoral muscle and liver were taken and cut into properly sized pieces (2 mm^3^) and fixed by immersion in buffered 3% glutaraldehyde in cacodylate buffer (pH 7.2) for at least 2 h at 4 °C. The tissue specimens were then post-fixed in 2% osmium tetroxide in cacodylate buffer (pH 7.2) for 1 h at 4 °C. Dehydration of the fixed tissues was performed using an ascending series of ethanol and then transferred into epoxy resin via propylene oxide [44]. Finally, the liver samples were embedded in a mixture of DDSA/NMA/EMbed-812 (Agar Scientific Ltd., London, UK). Ultra-thin sections (40–60 nm) were cut on a Reichert–Jung ultramicrotome and double stained with uranyl acetate and lead citrate. Evaluation of ultrastructure was performed using a transmission electron microscope TESLA BS-500 with Frame Transfer-1K-CCD-Camera (TRS, Mannheim, Germany).

### 2.8. Statistical Evaluation

Mean values in the analysed samples were processed using the PS IMAGO PRO 8.0 statistical package employing one-way analysis of variance (ANOVA). The normality of the data was checked with the Shapiro–Wilk test. The homogeneity of variance was also checked with the help of Levene’s test for homogeneity of variance. Tukey’s test was used to determine the significance of differences between the examined groups. The results were considered statistically significant when associated with a probability lower than 5%. The results with a probability lower than 1% were considered highly significant.

## 3. Results

The use of different forms of Se in the chickens’ diets at increased doses did not significantly (*p* > 0.05) affect the birds’ growth performance (Table 1). The results obtained are representative and do not differ from the expected performance for chicken [45].

The form of Se at the increased dose had no significant effect on the results of the slaughter analysis of chickens (Table 2). Final weight and carcass weight, slaughter yield and proportion of individual internal organs relative to BW were not significantly different from the CON group (*p* > 0.05).

In the present study, neither the supplemental form of Se nor the different doses of Se were found to have a negative effect on selected parameters of the chemical composition of the pectoral muscles (BM) of cockerels *(p* > 0.05) (Table 3). Only the total collagen content of the BM differed significantly (*p* ≤ 0.01). T2 pectoral muscles had the highest concentration of collagen in the muscle tissue, nearly 40% more collagen relative to CON. The addition of nano-Se did not significantly affect *(p* > 0.05) the collagen content of the BM with respect to the application of SS at 0.5 mg/kg feed.

The highest Se concentration (Figure 1A) in the pectoral muscles (CON = 0.27; T1 = 0.37; T2 = 0.42; T3 = 0.12 mg/kg) was determined in the chickens of the T2 group (*p* ≤ 0.01). For the liver (Figure 1B), the highest Se concentration in these tissues was found (*p* ≤ 0.01) in the CON group (CON = 3.79; T1 = 2.90; T2 = 0.86; T3 = 0.49 mg/kg).

The use of different forms of Se in the diet of chickens at a dose of 0.5 mg/kg feed had a significant effect (*p* ≤ 0.05) on the physico-chemical properties of breast muscle (Table 4). The BM of cockerels fed with SS (0.5 mg/kg feed) had the lowest pH_24_ (5.53) and the highest drip loss (3.01%) and WHC (3.11 cm^2^/g) relative to the analysed groups. Increasing the dose of Se in the form of other sources relative to CON significantly *(p* ≤ 0.05) affected WHC and drip loss parameters. However, the use of a dose of 0.5 mg/kg SS in the chickens’ diet had a significant effect on the deterioration of parameters related to the suitability of the meat for use in a specified product. The addition of SS at an increased dose had a significant effect (*p* ≤ 0.01) on higher levels of L* and b* parameters relative to the other test groups. This was also confirmed by analysis of the ΔE parameter. The muscles from cockerels with the increased dose of SS supplementation in the diet had the highest and visually apparent colour deviation (ΔE = 4.53) relative to the muscles from the CON.

Additionally, tissue sections of chicken livers and pectoral muscles from all four experimental groups were examined by transmission electron microscopy to determine the ultrastructural changes that correlate with the effects of the Se diet. The comparison of ultrastructural changes in pectoral muscle fibres is shown in Figure 2.

Ultrastructural analysis of the muscle fibres of broiler chickens fed a standard commercial diet containing inorganic Se at a level of 0.3 mg/kg BW of diet (CON) showed myofibrils with an irregular arrangement, becoming constricted and disrupted in places. Myofibrils are separated by sarcoplasm with visible mitochondria with slightly damaged structure. Sarcomeres showed regular organization of actin and myosin fibrils (Figure 2A,B). More ultrastructural changes in muscle fibres were found after supplementation with inorganic Se at a dose of 0.5 mg/kg BW. Irregularly arranged, markedly constricted myofibrils with a loose structure, patchy in places, separated by puffy sarcoplasm were seen. Also shown are swollen mitochondria with damaged cristae. Of note are sarcomeres with a significantly narrowed profile and damaged structure (Figure 2C,D). Figure 2E,F shows the ultrastructure of myofibrils of broiler chickens remaining dieting with an upper dose of Se in the organic form at a dose of 0.5 mg/kg feed. Normal parallel-running myofibrils with a compact regular structure are visible. The sarcomere structure with regular organization of actin and myosin filaments is well preserved. Mitochondria have the correct structure. The ultrastructure of the pectoral muscle of broiler chickens from the T3 group shows normal, parallel-running myofibrils with a compact, regular structure. Sarcomeres showed a regular organization of actin and myosin filaments. A proper mitochondrial profile with a slightly translucent mitochondrial matrix is shown (Figure 2G,H).

The ultrastructural analysis also showed statistically significant differences in sarcomere lengths (*p* ≤ 0.01). The shortest sarcomeres were found after supplementation with inorganic Se at a dose of 0.5 mg/kg BW, which was 1.64 µm compared to the control. The length of sarcomeres increased significantly in both groups of birds (T2 and T3) remaining on diets containing organic Se and nano-Se (Figure 3). In birds fed from the T2 group, the length of sarcomeres was 1.79 µm, and in birds from the T3 group, it was 1.81 µm.

The hepatocytes of broilers fed the diet containing inorganic Se at the level of 0.3 mg/kg feed (CON) show almost normal morphology (Figure 4A,B). Inside hepatocytes, a centrally located single nucleus of varying shape, usually spherical or oval, is visible. It is characterized by a conventional pattern of nuclear architecture: euchromatin is located mostly in the nuclear centre, whereas heterochromatin tends to be found in perinucleolar and perinuclear positions (the rim of heterochromatin typically lines the nuclear periphery) as well as forming random clumps throughout the nucleoplasm. The double membrane of the nuclear envelope (including perinuclear space and nuclear pores), is well visualized, and it is not affected (Figure 4A,B).

The cytoplasm is rich in normal, double-membraned mitochondria, whose inner membrane periodically invaginates into laminar cristae and surrounds a mitochondrial matrix of normal density (Figure 4B). The vast majority of mitochondria appear normal or show only little changes. In a few mitochondria, the presence of spherical electron-dense inclusions in the mitochondrial matrix is noteworthy (Figure 4A). A moderate amount of rough endoplasmic reticulum of normal organization forms a membrane network that surrounds mitochondria and extends across the entire cytoplasm. Ribosomes are bound to the cytoplasmic surfaces of cisternae in a more or less regular pattern. The characteristic lamellar membranous structure of the Golgi apparatus (ribbon-like shape) and its typical location near the nucleus can be seen. In addition, few primary lysosomes are visible near the nucleus (Figure 4A). Figure 4C,D presents the hepatocytes of the chicken fed the diet in the T1 group. Their morphology differs significantly from that described above. Although the cell nuclei have the correct structure, cytoplasmic organelles, especially mitochondria, are significantly altered. As shown in Figure 4C, almost all mitochondria exhibit varying degrees of hydropic degeneration; they have a vacuolated matrix and damaged inner membrane, including a high degree of cristae disorganization. In addition, the mitochondrial matrix of numerous mitochondria contains clusters of osmiophilic spherical deposits/inclusions (Figure 4D). Damaged mitochondria (*p* ≤ 0.01) represent 91.83% of the total amount in the cell (Figure 5). 

There is a noticeable reduction in the number of cisternae of the rough endoplasmic reticulum while maintaining its unique structure. The appearance of autophagic vacuoles (Figure 4C) and the reduction of primary lysosome numbers are noteworthy. 

The ultrastructure of hepatocytes of the chicken fed the diet with an upper dose of Se in the organic form (SY) at the level of 0.5 mg/kg feed (T2) appears normal (Figure 4E,F) and is similar to that observed in chickens fed the recommended diet. Mitochondria show no significant abnormalities in their morphology. However, some mitochondria are swollen and have damaged cristae. They account for 25.43% of the total amount in the cell (Figure 5). This is noteworthy that they do not contain any electron-dense material in the matrix. Instead, numerous primary lysosomes appeared (Figure 4F). They are distributed randomly within the cytoplasm. A nearly identical picture of hepatocytes was also observed in the chicken in group T3 (Figure 4G,H). Swollen mitochondria with damaged cristae also occur. They account for 26.96% of the total amount in the cell (Figure 5). 

The use of inorganic Se (CON and T1) in the chickens’ diet significantly (*p* ≤ 0.01) decreased AcP in pectoral muscle tissue. At the same time, HEX activity increased significantly *(p* ≤ 0.05) using SS in the chickens’ diet at an increased dose. The activity of aminopeptidases including AlaAP, LeuAP, ArgAP decreased in the breast muscle *(p* ≤ 0.01) when inorganic sources of Se were used (Table 5). The addition of other Se sources in the chickens’ diet other than SS influenced *(p* ≤ 0.05) the reduction of BGAL, BGLU and MAN activities. Nano-Se significantly *(p* ≤ 0.01) influenced the increase in aminopeptidase activity. In addition, AcP, HEX and aGlu activities increased significantly *(p* ≤ 0.05). 

Exposure of chicken to a diet with the upper dose of inorganic Se (T1) significantly *(p* ≤ 0.01) reduced the activity of AcP and HEX in the liver compared to the control group (CON). In contrast, BGRD and BGAL activities increased *(p* ≤ 0.01). As for aminopeptidase activity, LeuAP activity increased significantly, while AlaAP and ArgAP activity decreased *(p* ≤ 0.05). In general, in the liver, a decline in activity prevailed. Exposure of chicken to a diet with organic Se (T2) significantly increased AcP, BGAL and HEX activities and reduced BGRD activity *(p* ≤ 0.01). As for the activity of aminopeptidases, there was an increase in the activity of all of them. In general, in the liver of chickens in T2, the increase in activity prevailed. In the group of chickens fed a diet with nano-Se (T3), the activities of AcP, BGAL, HEX and all three estimated aminopeptidases increased significantly *(p* ≤ 0.01). Only BGRD activity decreased in this group of chickens *(p* ≤ 0.05). Overall, an increase in degradative activity prevailed, and changes in enzyme activity followed a similar pattern to the previous group, T2, fed a diet with organic Se. In blood serum, of all the enzymes analysed, only the activity of aminopeptidases changed statistically significantly in all the groups of chickens studied *(p* ≤ 0.01). A diet with the upper dose of inorganic Se significantly increased the activity of all aminopeptidases in serum, while feeding a diet with organic Se (T2) and nano-Se (T3) significantly decreased it.

The effect of Se type on the oxidative status in analysed tissues of broiler chickens was tested using the following determinations: vitamin C, glutathione (GSH), and 2,2-diphenyl-1-picrylhydrazyl (DPPH). In the pectoral muscle of chickens fed the upper dose of inorganic Se (T1 group), all indicators decreased, while in fed organic Se (T2 group) and nano-Se (T3), all indicators increased significantly. In the liver, both groups of chickens: T2 and T3, obtained significantly higher vitamin C, DPPH and glutathione results than the group considered a control. In contrast, the T1 group had statistically significantly lower vitamin C and glutathione levels and a higher result for DPPH than the CON group. Analysis of serum antioxidant potential in the T1 group showed a statistically significant reduction in vitamin C and DPPH levels *(p* ≤ 0.05), while glutathione level increased *(p* ≤ 0.05). Serum levels of all analysed factors decreased in chickens fed a diet supplemented with organic Se. The addition of nano-Se to the diet decreased the levels of vitamin C and GSH and increased the level of DPPH *(p* ≤ 0.05).

## 4. Discussion

Food can be supplemented with selenium (Se) in various forms, such as inorganic, organic, and nanoparticle forms. The metabolism of these forms is different in birds. The chemical form and concentration of Se have significant roles in the rate of absorption, retention, and metabolism. Most inorganic Se is excreted in the urine, while nano-Se particles are excreted in faeces [46]. It has been shown that the inorganic form of Se has low bioavailability, accelerates oxidation processes and can be toxic, especially at high concentrations. In contrast, organic-Se and nano-Se exhibit low toxicity, high adsorption capacity, high bioavailability and high catalytic efficiency in chickens, sheep and goats [46,47,48]. Selenium is an important trace element that upregulates a vital component of the antioxidant defiance mechanism by controlling the body’s glutathione pool and Se-dependent antioxidant enzymes such as superoxide dismutase and glutathione peroxidase [49,50]. These enzymes can help in reducing the concentration of hydrogen peroxide and lipid peroxides and enhance the immune response in numerous species of animals [51]. Several studies have illustrated that the dietary Se form influences growth performance, meat quality characteristics and antioxidative properties in chickens [52,53]. 

Based on the study, the different forms of Se in the feed at an increased dose (0.5 mg/kg) were not found to negatively affect the growth performance of the chickens. Our results showed that the addition of nanoparticle Se did not adversely impact the growth performance of chickens, indicating that nano-Se has high bioavailability and low toxicity compared to inorganic forms of Se such as SS [54]. The results obtained in this study are consistent with those of Yoon et al. [55], which found that the source of Se in the feed did not affect the growth performance of chickens. In contrast, groups of chickens fed a lower dose of 0.2 mg/kg of dietary organic Se or nano-Se had similar growth performance compared to a group supplemented with the same level of Se in the form of selenite [56]. In a study by Ahmad et al. [36], it was found that the application of increased Se intake (0.4 mg/kg feed) in the SY form and nano-Se was able to cover the requirement of this element by the birds and provide optimal growth conditions with no differences between the application of SY and nano-Se. There are known reports where Se influences the growth performance of birds, which is related to the expression of selenoprotein P and type I selenoenzymes, which play a key role in the synthesis of thyroid hormones and Se transport, both of which contribute to the proper functioning and growth of avian organisms [57]. 

The use of different sources of Se at an increased dose, as in the case of chicken growth performance, had no significant effect *(p* ≥ 0.05) on the slaughter analysis results of cockerels at day 42 of rearing. Higher slaughter and BM performance was found in broilers that received SY in the feed. In contrast, Payne and Southern [58], in an experiment conducted on chickens receiving SS and SY, found that slaughter performance was not dependent on the form of Se in the feed. Downs et al. [52,58] found a similar relationship in their study on the use of SY in broiler feeding. The non-significant difference in body weight and other slaughter parameters in the present study may be due to feeding a balanced diet with appropriate practice [59] and optimal microclimate conditions with the welfare of the birds unaffected by, among other things, heat stress or other factors that may negatively affect the rearing performance of the chickens. 

The highest concentration of Se was found in the BM of T2 cockerels, whereas the highest concentration of Se was found in the liver of the T1 group (SS, 0.5 mg/kg). This result is different from what was reported in [60]. Studies have indicated that nano-Se has a higher bioavailability and a lower risk of toxicity than other forms of Se [61]. Based on the results of the present study, it seems reasonable to recommend that broilers take nano-Se at a dose of 0.3–0.5 mg/kg [22]. The organic form of Se is accumulated to a greater extent than the inorganic form [62,63]; presumably, this may be related to differences in metabolic pathways between the inorganic and organic forms of Se. Inorganic Se is absorbed through the intestine through a passive diffusion process, with organic Se through an active transport mechanism [62]. A final product with increased nutritional value can be achieved by adequate supplementation of nano-Se at an increased dose (0.5 mg/kg). Our study confirms that nano-Se and liver enriched with nano-Se in this way are safe and can provide a valuable source of Se [34].

The post-mortem transformation of chicken muscle leads to an accumulation of lactic acid, which directly leads to a reduction in tissue pH. The delayed reduction in pH post-mortem results in reduced protein denaturation, which translates into an improved ability of skeletal muscle to retain its own water [64]. In the present study, a significant effect of Se application at an increased dose on pH_24_ was found. BM of cockerels in which SS was used in the diet at a dose of 0.5 mg/kg feed had the lowest pH_24_ and thus higher drip loss (3.01) and WHC (3.11), while the differences between CON, SY and nano-Se were not significant. This helps to conclude that the use of other forms of Se in chicken feed versus SS at an increased ration has a positive effect on the physico-chemical properties related to water retention in BM. Meat colour is dependent on a number of factors, including pH, myoglobin concentration, nitrite, etc. [65]. Se can significantly improve serum glutathione peroxidase activity, increase the resistance to oxidation of myoglobin or oxymyoglobin, among others, and deepen the colour of meat [22,66]. In this study, an increased dose of Se SS increases lightness (L*) and yellowness (b*), which directly affect the consumer’s visual assessment of the pectoral muscle (ΔE).

Ultrastructural studies conducted showed the most significant changes in the muscle fibres of broiler chickens after supplementation of the inorganic Se at a dose of 0.5 mg/kg diet (Figure 2C,D). We showed a change in the profile of myofibrils and damage to sarcomeres and mitochondria. Furthermore, measurements of sarcomere size showed a reduction in length to 1.64 µm, which may suggest reduced elasticity of muscle tissue. This may be due to the weaker uptake of Se, which is more difficult to bind to glutathione peroxidase, which is responsible for scavenging oxygen free radicals and preventing oxidative stress [67,68,69]. In contrast, chickens that were fed Se in organic form (T2) and Se in nano form (T3) showed a slightly different picture of ultrastructural changes (Figure 2E–H). In this case, a stable structure of myofibrils with sarcomeres with a regular profile of actin and myosin fibres was demonstrated. No significant changes in mitochondrial structure were shown. However, measurements of the size of the sarcomeres showed an increase in length to 1.79 µm after organic Se (T2), and an increase to 1.81 µm after nanoparticles, the opposite of the group receiving inorganic Se. According to data [70], the length of sarcomeres can influence the better elasticity of muscle tissue and thus the quality of broiler chicken meat. To reveal the effect of different sources of Se supplementation on the ultrastructure of hepatocytes, the ultra-thin sections of the liver of chicken from all experimental groups (CON, T1, T2, T3) were also analysed by transmission electron microscopy (TEM). Our research has shown that the primary effect of Se at the ultrastructural level of the liver was the appearance of clusters of electron-dense (osmiophilic) inclusions within the mitochondrial matrix. However, these pathological changes were mainly observed in the hepatocytes of the chickens from the T1 group (Figure 4C,D) and to a minor extent in the mitochondria of the chickens fed the recommended diet containing SS at the level of 0.3 mg/kg feed (Figure 4A,B). Inclusions were not observed in the hepatocytes of chickens (T1) of the other groups, T2 and T3. In addition, almost all mitochondria in hepatocytes of chickens from the T1 group exhibit varying degrees of hydropic degeneration. They were swollen and had damaged inner membranes, including a high degree of cristae disorganization. The inclusions mentioned early on were found in the mitochondrial matrix that was transparent due to the influx of water. Abundant, spherical shape electron-dense deposits in the mitochondrial matrix were seen in numerous mitochondria of chicken fed with the upper dose of inorganic Se (Figure 4D). Abnormal mitochondria represent 91.83% of the total amount (Figure 5). In our opinion, hepatocyte mitochondria of chicken from the T1 group show features of a toxic response to excess inorganic Se in the feed. The effect of changes in the matrix volume in the physiological range is to stimulate the electron transfer chain and oxidative phosphorylation in order to satisfy the metabolic requirements of the cell. However, excessive matrix swelling caused by the continued opening of mitochondrial permeability transition pores (*p*TP) and other PTP-independent mechanisms makes mitochondrial function and integrity less efficient and leads to cell death [71]. Similar observations regarding the accumulation of electron-dense inclusions in the mitochondrial matrix were made by Medina et al. [72], who studied the effects of Se (Na_2_SeO_3_; SS) on the growth of three cell lines cultured in vitro. They performed analyses by transmission electron microscopy to determine any ultrastructural changes which correlated with the effects of Se treatment. They observed that the vast majority of the mitochondrial matrices were filled with dense osmophilic deposits. The accumulation was dependent on the dose of Se in the culture medium, the time of exposure, and the cell line. However, Se did not alter cytoplasmic microtubules or intermediate filament networks because they obtained abundant desmosomes [72]. Furthermore, they examined the chemical nature of the electron-dense material in the mitochondrial matrix by X-ray microanalysis of sections of studied cells. They obtained, as a result, an increased level of calcium, iron, and Se in the mitochondrial matrix [72]. There was some speculation about the nature of the material accumulated, as the nature of the mitochondrial inclusion was generally unknown, although it could be protein complexed by Se since Se reacts easily with mercapto groups of organic compounds [72,73]. Medina et al. [72] suggested that whereas high doses of Se inhibit the growth of cells, low doses stimulate the growth of some cell populations. They speculated that one of the early reasons of Se-mediated growth inhibition may be a modulation of mitochondrial function [72]. Klug et al. [73] found that succinic dehydrogenase was inhibited in vivo by sodium selenite. Liu et al. [74] used TEM to detect the accumulation of lentinan (LNT)-functionalized nano-Se in the mitochondria of tumour cells, and furthermore, they investigated the mechanism of selene targeting mitochondria. Analysis of ultra-thin sections showed that Se enters the cells through the caveolae-mediated endocytosis pathway and then enters the mitochondria via mitochondrial membrane fusion [74]. Zahedi et al. [75] analysed the effects of Se tetrachloride (SeCl_4_) on the mitochondria of lung A549 cells. Analysis of mitochondrial morphology showed a significant increase in the swollen phenotype and a decrease in mitochondrial motility in Se treated cells [75]. Furthermore, they performed double staining of the cells with autophagy marker GFP-LC3 and MitoTracker. The results of this staining showed that the swollen mitochondria co-labeled with autophagosomes, indicating their targeted degradation via mitophagy [75]. Noteworthy in our study is the simultaneous appearance of damaged mitochondria and autophagic vacuoles, as well as a reduction in the number of primary lysosomes in hepatocytes of chickens treated with a high dose of inorganic Se (T1). In hepatocytes from chickens fed the upper dose of organic Se (T2) and nano-Se (T3), in contrast to the T2 group, only some mitochondria were swollen and had damaged cristae (accounting for 25.43% and 26.96% of the total amount in the cell, respectively). In contrast, numerous primary lysosomes appeared (Figure 4F). Overall, in the groups of chickens (CON, T2 and T3), in contrast to the T1 group, no significant pathological changes were observed in the ultrastructure of hepatocytes, either with respect to mitochondria or other organelles. 

Parallel to morphological analysis, the activity of degradative enzymes was estimated. Morphological images correlate with the results of degradative enzyme activity (Table 5). In the liver of chickens exposed to a diet with an upper dose of inorganic Se, a decrease in the activity of the analysed enzymes prevailed. This indicates that the enzymes were involved in intracellular digestion processes, as evidenced by the presence of autophagic vacuoles and depletion of primary lysosomes, which are a cellular reservoir of acid hydrolases (Figure 4C,D). In contrast, an increase in the activity of the analysed enzymes in the liver prevailed in the other two groups of chickens (exposed to a diet with organic Se—T2, and nano-Se—T3), indicating a reduction in the level of lysosomal degradation. The ultrastructure of hepatocytes of both groups showed numerous primary lysosomes, presumably due to low lysosome consumption. The decrease in enzyme activity in the T1 group may indicate an increase in the degradation of organelles damaged by an excessive dietary supply of inorganic Se. According to Marzella et al. [76], initially, the enzyme activity increases in parallel with the induction of degradation, but it later decreases with the advanced stage of the sequestered cell organelle degradation [77,78]. With more advanced stages of lysosomal degradation, the activity seems to decline due to enzyme reserves consumption [76,77] and possibly insufficient synthesis. According to Kalamida et al. [79], the increase in lysosome accumulation, presumably due to a low rate of lysosome consumption, is accompanied by an increase in cathepsin D expression. Zahedi et al. [75], analysing the effects of Se tetrachloride on the mitochondria of lung A549 cells, obtained a significant dose-dependent increase in mitochondrial protein oxidation [75]. Several research teams confirm that Se (SS, selene-lentinan-functionalized, nano-Se) has been implicated in mitochondrial toxicity, with reactive oxygen species playing an important role [74,75,80,81]. They confirmed that after Se treatment, mitochondria become the main organelles of ROS production. Sun et al. [81] showed that excessive Se can enhance the toxicity of other metals (such as arsenic) by reacting with S–adenosylmethionine and glutathione. Mitochondria are very dynamic structures and have been used to study a variety of biological problems such as stress and drug response [75]. Environmental toxicants can have a negative impact on mitochondria and alter their morphology [75,82]. It is well known that mitochondrial dysfunction is a prominent phenomenon in the pathogenesis of a variety of diseases. In summary, mitochondria appear to be the main targets of the upper dose of Se, and changes in the structure and function of this organelle are commonly observed in Se-treated cells [72,74,75,83].

The influence of Se on the oxidative status in the analysed tissues of broiler chickens was tested using the following determinations: vitamin C, glutathione (GSH), and 2,2-diphenyl-1-picrylhydrazyl (DPPH). Both groups of chickens in T2 and T3 obtained significantly higher vitamin C, DPPH and glutathione results in muscle and liver tissues than in the group considered a control. In contrast, group T1 obtained significantly lower results in these tissues. The increase in the levels of antioxidant indicators in the muscle and liver tissues of chickens from the T2 and T3 groups indicates better oxidative stability of the organism, in contrast to the T1 group, where we obtained lower parameters. Visha et al. [84] obtained very similar results; namely, birds supplemented with the nano- and organic Se showed higher total antioxidant capacity in the serum and tissues compared to the inorganic Se supplemented group. Se is known to play an essential role in protecting cells from oxidative damage by affecting the antioxidant levels and activities of selenoenzymes [85]. Nano-Se is capable of scavenging free radicals by improving the activity of seleno-enzymes [85,86] and growth improvement along with the status of serum oxidants and retention of selenium in vivo. In comparison with other selenium species such as SeMet, SeCys, and SY, nano-Se showed lower acute toxicity but increased the activities of selenoenzymes. The antioxidant effect of nano-Se is mainly associated with glutathione peroxidase family (GPXs) and thioredoxin reductase (TR). The GPXs have a capacity to detoxify an extended range of peroxides, such as H_2_O_2_, phospholipid hydroperoxide, fatty acid hydroperoxides, and hydroperoxyl groups of thymine [87]. In conclusion, the study concluded that the use of nano-Se in the diet can be effective in increasing the antioxidant potential in the organisms of chickens, thereby increasing the ability to detoxify a wide range of peroxides.

## 5. Conclusions

Based on this study, the use of different forms of Se in broiler chicken diet besides commonly used sodium selenate had a positive effect on the physico-chemical properties of the breast muscle and on the health status, antioxidant potential and ultrastructure of the breast muscle and liver. It also improved quality parameters of the pectoral muscle, such as the length of the sarcomeres, which resulted in better elasticity of the muscle tissue and thus in meat quality. In addition, the use of nano-Se at an increased dose was confirmed to protect against mitochondrial damage in hepatocytes and increased antioxidant potential. Nano-Se did not show degenerative and toxic effects similar to sodium selenite at a dose of 0.5 mg/kg feed. The use of nano-Se at a dose higher than the recommended dose had high bioavailability and low toxicity without negatively affecting the growth performance of chickens.

## Figures and Tables

**Figure 1 antioxidants-12-00905-f001:**
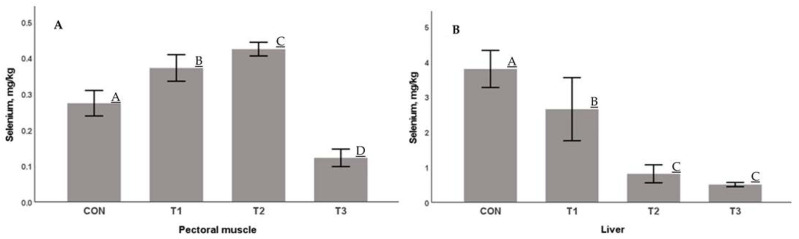
The effect of using increased doses of various forms of Se on Se content in tissues: (**A**) breast muscle; (**B**) liver. Data are given as mean ± SEM (n = 10). ^A, B, C, D^—values denoted with various letters differ significantly at *p* ≤ 0.01; CON—control group; T1—diet with upper dose of inorganic Se (0.5 mg/kg feed); T2—diet with upper dose of Se in the organic form (0.5 mg/kg feed); T3—diet with upper dose of Se in the form of nanoparticles (0.5 mg/kg feed).

**Figure 2 antioxidants-12-00905-f002:**
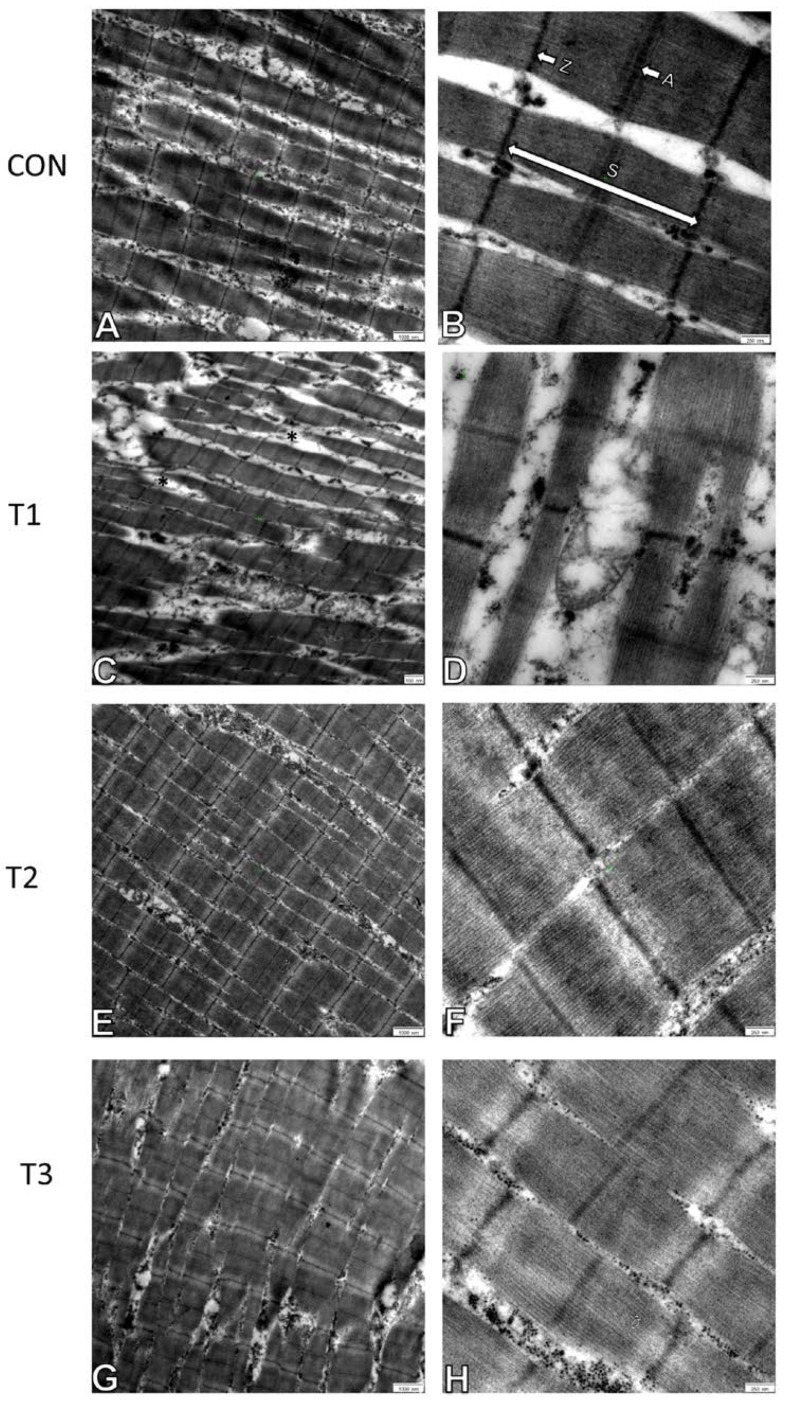
Ultrastructure of representative myofibers of the pectoral muscle of cockerels: CON—control group (**A**,**B**), T1—diet with upper dose of inorganic Se (**C**,**D**), T2—diet with upper dose of Se in the organic form (**E**,**F**), T3—diet with upper dose of Se in the form of nanoparticles (**G**,**H**). Scale bar 1000 nm (**A**,**C**,**E**,**G**) and 250 nm (**B**,**D**,**F**,**H**). S—sarcomere; A—A band; Z—Z band; *—indicates the disruption of myofibers.

**Figure 3 antioxidants-12-00905-f003:**
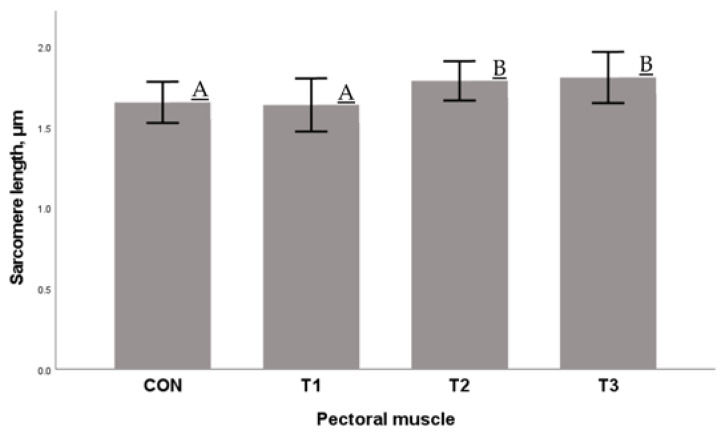
Length of sarcomeres in the pectoral muscle of cockerels. Data are given as mean ± SEM (n = 10); ^A, B^—values denoted with various letters differ significantly at *p* ≤ 0.01; CON—control group; T1—diet with upper dose of inorganic Se (0.5 mg/kg feed); T2—diet with upper dose of Se in the organic form (0.5 mg/kg feed); T3—diet with upper dose of Se in the form of nanoparticles (0.5 mg/kg feed).

**Figure 4 antioxidants-12-00905-f004:**
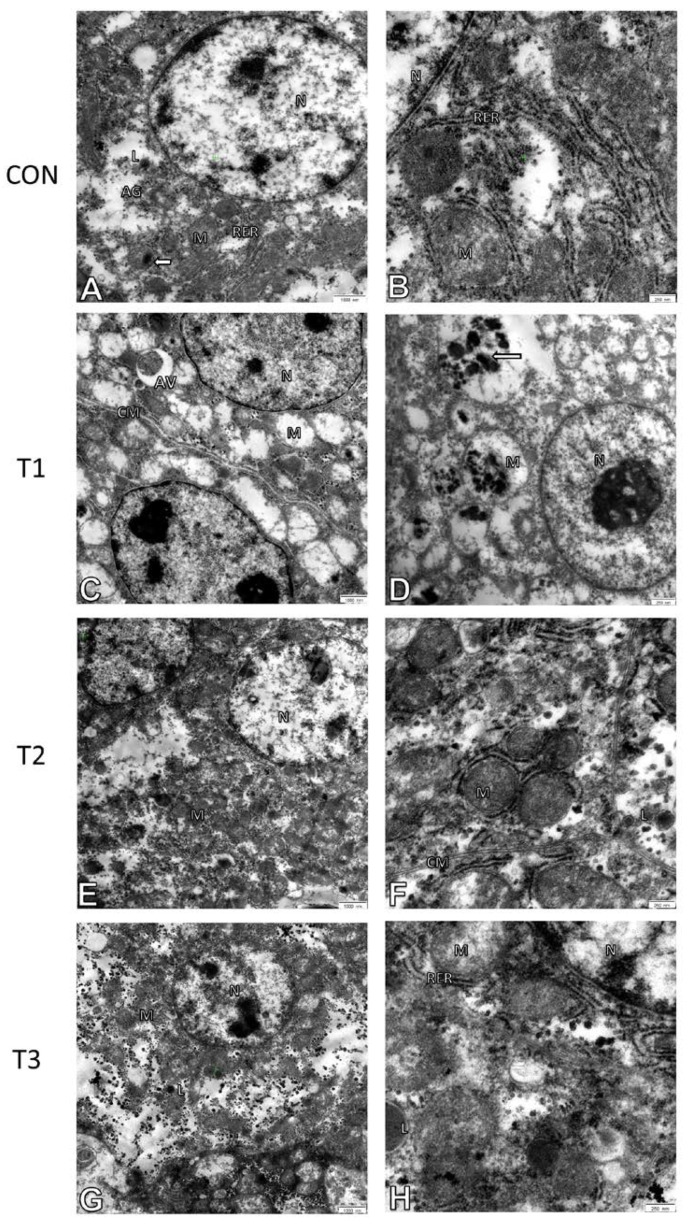
Ultrastructure of representative hepatocytes of cockerels: CON—control group (**A**,**B**), T1—diet with upper dose of inorganic Se (**C**,**D**), T2—diet with upper dose of Se in the organic form (**E**,**F**), T3—diet with upper dose of Se in the form of nanoparticles (**G**,**H**). Scale bar 1000 nm (**A**,**C**,**E**,**G**) and 250 nm (**B**,**D**,**F**,**H**). N—nucleus; M—mitochondria; RER—rough endoplasmic reticulum; L—lysosomes; AV—autophagic vacuole; CM—cell membrane; arrow—electron dense deposits in mitochondria.

**Figure 5 antioxidants-12-00905-f005:**
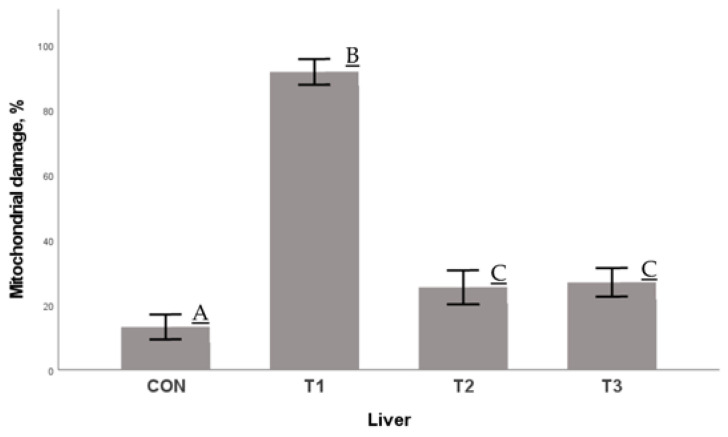
Mitochondrial damage in the liver. Data are given as mean ± SEM (n = 10); ^A, B, C^—values denoted with various letters differ significantly at *p* ≤ 0.01; CON—control group; T1—diet with upper dose of inorganic Se (0.5 mg/kg feed); T2—diet with upper dose of Se in the organic form (0.5 mg/kg feed); T3—diet with upper dose of Se in the form of nanoparticles (0.5 mg/kg feed).

**Table 1 antioxidants-12-00905-t001:** Chicken growing results.

Indices	Group				SEM	*p* Value
	CON	T1	T2	T3		
Body weight (BW), g:						
1 day	37.32	37.87	38.24	38.18	0.167	0.415
42 day	3012.22	2999.30	3037.50	2989.70	22.728	0.554
FCR, kg kg^−1^	1.63	1.55	1.57	1.53	0.023	0.130
Mortality, %	2.66	1.33	2.66	1.33	0.012	0.479

Data are given as mean ± SEM (n = 10); CON—control group; T1—diet with upper dose of inorganic Se (0.5 mg/kg feed); T2—diet with upper dose of Se in the organic form (0.5 mg/kg feed); T3—diet with upper dose of Se in the form of nanoparticles (0.5 mg/kg feed).

**Table 2 antioxidants-12-00905-t002:** Results of male broiler chicken slaughter analysis.

Indices	Group				SEM	*p* Value
	CON	T1	T2	T3		
Body weight, g	3343.75	3235.44	3251.76	3236.00	19.964	0.166
Carcass weight, g	2525.60	2446.22	2438.46	2417.74	17.626	0.143
Dressing percentage, g per 100 g BW	75.56	75.60	75.00	74.71	3.293	0.738
Breast muscles, g per 100 g BW	28.19	28.42	28.00	28.20	1.484	0.811
Leg muscles, g per 100 g BW	19.38	19.77	18.94	19.78	3.212	0.785
Gizzard, g per 100 g BW	0.82	0.83	0.83	0.83	0.023	0.710
Heart, g per 100 g BW	0.75	0.79	0.80	0.80	0.014	0.620
Liver, g per 100 g BW	2.16	2.29	2.29	2.33	0.042	0.380
Abdominal fat, g per 100 g BW	1.48	1.49	1.51	1.54	0.023	0.810

Data are given as mean ± SEM (n = 10); CON—control group; T1—diet with upper dose of inorganic Se (0.5 mg/kg feed); T2—diet with upper dose of Se in the organic form (0.5 mg/kg feed); T3—diet with upper dose of Se in the form of nanoparticles (0.5 mg/kg feed).

**Table 3 antioxidants-12-00905-t003:** Selected chemical components of breast muscles of male broiler chickens.

Indices, %	Group				SEM	*p* Value
	CON	T1	T2	T3		
Moisture	75.32	76.41	75.46	74.86	0.254	0.065
Total fat	0.87	0.89	1.48	1.10	0.243	0.121
Total protein	23.53	23.32	22.85	23.35	0.194	0.383
Total ash	1.40	1.41	1.44	1.50	0.198	0.175
Total collagen	0.37 ^A^	0.44 ^AB^	0.93 ^C^	0.75 ^BC^	0.129	<0.001

Data are given as mean ± SEM (n = 10); ^A, B, C^—values denoted with various letters differ significantly at *p* ≤ 0.01; CON—control group; T1—diet with upper dose of inorganic Se (0.5 mg/kg feed); T2—diet with upper dose of Se in the organic form (0.5 mg/kg feed); T3—diet with upper dose of Se in the form of nanoparticles (0.5 mg/kg feed).

**Table 4 antioxidants-12-00905-t004:** Physico-chemical properties of breast muscles of male broiler chickens.

Indices	Group				EM	*p* Value
	CON	T1	T2	T3		
pH_24_	5.64 ^A^	5.53 ^C^	5.73 ^B^	5.72 ^AB^	0.017	<0.001
Drip loss, %	2.81 ^A^	3.01 ^B^	2.57 ^A^	2.51 ^A^	0.212	0.023
WHC, cm^2^/g	2.34 ^A^	3.11 ^B^	2.70 ^A^	2.54 ^A^	0.123	0.016
L* lightness	63.70 ^A^	67.46 ^B^	61.61 ^A^	60.71 ^A^	0.572	<0.001
a* redness	12.50	12.71	12.73	12.53	0.161	0.943
b* yellowness	9.69 ^A^	12.20 ^B^	9.16 ^AC^	8.11 ^C^	0.294	<0.001
ΔE CON: T1–T3	0.00	4.53	2.17	3.38	-	-

Data are given as mean ± SEM (n = 10). ^A, B, C^—values denoted with various letters differ significantly at *p* ≤ 0.01; parameter L* (colour brightness) can have values from 0 to 100. Parameters a* (redness) and b*(yellowness) are tri-chromaticity coordinates and can have positive and negative values: +a* corresponds to red, +b* to yellow. ΔE—absolute colour difference; ^A, B, C^—values denoted with various letters differ significantly at *p* ≤ 0.01; CON—control group; T1—diet with upper dose of inorganic Se (0.5 mg/kg feed); T2—diet with upper dose of Se in the organic form (0.5 mg/kg feed); T3—diet with upper dose of Se in the form of nanoparticles (0.5 mg/kg feed).

**Table 5 antioxidants-12-00905-t005:** Enzymatic activity and antioxidant potential in selected tissues of male broiler chickens.

Indices	Group																	
	Breast Muscle						Liver						Serum					
	CON	T1	T2	T3	SEM	*p* Value	CON	T1	T2	T3	SEM	*p* Value	CON	T1	T2	T3	SEM	*p* Value
AlaAP, nmol/mg protein/h	208.64 ^A^	171.07 ^A^	274.29 ^B^	287.73 ^B^	11.529	<0.001	265.91 ^A^	267.32 ^A^	406.13 ^B^	467.25 ^B^	23.211	0.001	40.12 ^A^	46.32 ^A^	22.17 ^B^	28.91 ^B^	2.0630	<0.001
LeuAP, nmol/mg protein/h	132.47 ^A^	129.74 ^A^	160.09 ^AB^	189.48 ^B^	6.335	<0.001	389.71 ^A^	366.43 ^A^	407.05 ^AB^	524.87 ^B^	19.878	0.013	48.38 ^AB^	51.22 ^A^	29.40 ^C^	40.71 ^B^	1.928	<0.001
ArgAP, nmol/mg protein/h	185.15 ^A^	172.23 ^A^	223.30 ^B^	249.21 ^C^	6.758	<0.001	319.19 ^A^	321.11 ^A^	369.28 ^AB^	414.11 ^B^	11.137	0.001	32.06 ^A^	34.41 ^A^	16.31 ^B^	23.45 ^B^	1.614	<0.001
AcP, nmol/mg protein/h	1330.75 ^A^	1538.31 ^A^	1899.71 ^B^	2137.26 ^B^	71.710	<0.001	1528.39 ^A^	1240.78 ^B^	1568.14 ^A^	1759.12 ^C^	42.830	<0.001	31.63	35.69	37.30	30.83	2.795	0.831
BGDR, nmol/mg protein/h	129.17	125.84	129.88	137.27	4.114	0.816	347.44 ^A^	427.73 ^B^	318.99 ^A^	300.51 ^A^	13.431	0.001	9.27	9.02	10.55	8.49	1.066	0.926
BGAL, nmol/mg protein/h	429.23 ^b^	400.83 ^ab^	372.47 ^a^	391.54 ^ab^	7.307	0.036	353.88 ^A^	357.39 ^A^	381.43 ^A^	483.94 ^B^	14.620	0.001	8.12	9.58	12.02	9.73	0.866	0.477
BGLU, nmol/mg protein/h	237.81 ^AB^	264.24 ^B^	211.54 ^A^	237.58 ^AB^	5.206	0.001	383.20	401.92	380.35	358.56	12.127	0.686	9.39	7.95	10.20	8.00	1.085	0.868
HEX, nmol/mg protein/h	617.80 ^a^	692.00 ^ab^	659.41 ^ab^	745.95 ^b^	17.003	0.042	1064.01 ^AB^	941.48 ^A^	1114.53 ^AB^	1212.82 ^B^	30.675	0.008	58.65	56.58	50.85	52.79	3.913	0.807
aGlu, nmol/mg protein/h	76.24 ^A^	71.76 ^A^	87.50 ^AB^	102.46 ^B^	3.328	0.001	323.77	240.78	304.10	294.70	12.844	0.093	7.26	6.65	8.22	6.45	0.746	0.852
MAN, nmol/mg protein/h	327.70 ^A^	328.87 ^A^	287.92 ^B^	292.97 ^B^	5.160	0.001	381.34	306.85	358.08	372.77	13.256	0.192	6.39	7.73	12.32	8.22	0.984	0.166
Vit. C, mg/100 mL	1.89 ^A^	1.80 ^A^	4.70 ^B^	3.15 ^AB^	0.323	0.001	2.57 ^AB^	2.02 ^A^	4.16 ^BC^	4.36 ^C^	0.287	0.001	4.46 ^ab^	3.85 ^b^	3.75 ^b^	6.73 ^a^	0.381	0.011
DPPH, %	52.55 ^AB^	51.49 ^A^	55.83 ^BC^	56.12 ^C^	0.588	0.002	85.53 ^A^	85.71 ^AB^	86.42 ^BC^	86.56 ^C^	0.132	0.004	82.82 ^A^	81.84 ^AB^	74.27 ^BC^	71.20 ^C^	1.369	0.002
GSH mM-SH	0.05 ^A^	0.04 ^A^	0.20 ^B^	0.25 ^B^	0.021	<0.001	0.85 ^AB^	0.71 ^A^	0.89 ^B^	0.95 ^B^	0.026	0.004	0.99 ^A^	1.00 ^A^	0.98 ^A^	1.28 ^B^	0.034	0.001

Data are given as mean ± SEM (n = 10); ^A, B, C^—values denoted with various letters differ significantly at *p* ≤ 0.01; ^a, b^—values denoted with various letters differ significantly at *p* ≤ 0.05; SEM—standard error of mean; CON—control group; T1—diet with upper dose of inorganic Se (0.5 mg/kg feed); T2—diet with upper dose of Se in the organic form (0.5 mg/kg feed); T3—diet with upper dose of Se in the form of nanoparticles (0.5 mg/kg feed). AlaAP—alanine aminopeptidase; LeuAP—leucine aminopeptidase; ArgAP—arginineaminopeptidase; AcP—acid phosphatase; BGRD—beta-glucuronidase; BGAL—beta-galactosidase; BGLU—beta-glucosidase; aGlu—alpha-glucosidase; MAN—mannosidase; HEX—N-acetyl-BD-hexosaminidase; vit. C—vitamin C; GSH—glutathione; DPPH—2;2-diphenyl-1-picrylhydrazyl.

## Data Availability

All data are contained within the article.

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
