# Peer review of "Nano-Sized Selenium Maintains Performance and Improves Health Status and Antioxidant Potential While Not Compromising Ultrastructure of Breast Muscle and Liver in Chickens"

_antioxidants, 2023, doi:10.3390/antiox12040905_

Round 1

Reviewer 1 Report

Authors write on lines 41-42 that selenium is a component of proteins. This is not the correct phrase for a trace element.

Second, lipid oxidation with MDA and protein carbonills have not been evaluated,

Conclusions need totally re-writing because authors say that organic forms of Selenium imporves performance but they declare no significant diferrences.

Authors must explain unbieased the role of inorganic and organic selenium. The organic form cannot totally replace the inorganic one. Do they know this information?

Recent references are needed.

Author Response

Dear Reviewer 1,

The authors are very grateful for valuable comments to improve our paper. We have implemented all the remarks indicated and believe that the paper is now suitable for publication. To support visibility of the modifications made, we have introduced changes in the text by highlighting them in yellow within the manuscript.

  1. Authors write on lines 41-42 that selenium is a component of proteins. This is not the correct phrase for a trace element.

Answer: This piece of text has been corrected: “Selenium is an essential mineral element with important biological functions for the whole body through incorporation into at least 30 selenoproteins”, please see the line 41-42.

  1. Second, lipid oxidation with MDA and protein carbonills have not been evaluated?

Answer: Unfortunately, these indices were not determined in the present study 

  1. Conclusions need totally re-writing because authors say that organic forms of Selenium imporves performance but they declare no significant diferrences.

Answer: The chapter 'Conclusions' has been improved, please see line 627-637.

  1. Authors must explain unbieased the role of inorganic and organic selenium. The organic form cannot totally replace the inorganic one. Do they know this information?

Answer: This piece of text has been corrected: “The effect of selenium on the health of birds depends on the form in which it is sup-plied to the diet. Inorganic Se is passively absorbed from the intestine through a simple diffusion process and competes with many mineral elements for absorption pathways and organic Se is actively absorbed through amino acid transport mechanism and has a higher bioavailability than the inorganic form [11]. The combination of SeMet + sodium selenite is more efficient than their individual treatments for Se deposition in egg and chicken embryo tissues. Although many studies have shown that organic Se is easier to absorb than inorganic Se, the reduction of competitive absorption leads to the higher absorption efficiency and production performance of the combination of the two Se sources [12]. A thorough understanding of this mechanism requires further research and analysis”, please see the line 49-59.

  1. Recent references are needed.

Answer: Due to the addition of selenium information in the chapter: "Introduction", new references have been added.

Reviewer 2 Report

Please see the comments in the attached file.

Author Response

Dear Reviewer 2,

The authors are very grateful for valuable comments to improve our paper. We have implemented all the remarks indicated and believe that the paper is now suitable for publication. To support visibility of the modifications made, we have introduced changes in the text by highlighting them in yellow within the manuscript.

  1. Abstract: I suggest that the authors of the manuscript significantly improve the information presented. This must be done to attract more attention of readers to research on this scientific issue.

Answer: The abstract has been corrected, please see line 20-36.

  1. Keywords I suggest adding a few terms to expand the search (this is a recommendation)

Answer: Revised the Abstract and added new keywords, please see line 37-38

  1. Introduction: I suggest that the authors significantly improve, expand the search for scientific articles on this topic, perform a comprehensive analysis, focus on the problems of the scientific direction and its significance for further own research. I did not like the brevity of this section, not the in-depth analysis of publications in peer-reviewed scientific journals. A lot of scientific studies on the use of selenium nanoparticles in poultry feeding have been published. Make and Add this analysis

Answer: Chapter revised: Introduction, please see line 41-116.

  1. Materials and Methods:
    1. During what period were the studies carried out?

Answer: We specified the period of experiment (from hatching day till 42 days of life), please see line 121.

  1. This section does not feature Selenium. It is necessary to add a full description of the Selenium preparations used in this experiment.

Answer: Thank you for these remarks, we provided additional information regarding used Se forms, please see lines 136-144.

  1. What rules (ethical standards) did you use for working with broiler chickens? Please indicate.

Answer: The study was conducted in accordance with the guidelines established by the European Union and Polish Law on Animal Protection. All detailed description of study-protocol is provided now in the section “Institutional Review Board Statement”, please see lines 648-663.  

  1. Animals and Diets: How was the feed mixture prepared with selenium for feeding poultry?

Answer: All diets were prepared according to protocol used in our laboratory. A one batch of based diet was obtained from the feed mill (this diet did not contain vitamin-mineral premix). Subsequently, it was divided in four sub-batch (CON, T1, T2, T3 and T4), and each of sub-batch was supplemented with vitamin-mineral premix contained respective form and dose of Se. Eventually, each of sub-batch was mixed with respective vitamin-mineral premix to ensure homogeneity.

  1. Sampling procedures: Why were only males selected for the experiment?

Answer: The main goal of this was that that by using one sex it is possible to reduce the numbers of birds used in the experiment, and it allows reducing cost of analyses (RRR-rule).

  1. Sampling procedures: I ask the co-authors of the manuscript to indicate the methodology in this section.

Answer: Corrected, please see line 148-150.

  1. Selenium content: Determination of Se content in breast muscle (n=10) and liver (n=10) was performed according to the Bień et al.[6].

Answer: Corrected, please see line 162-165.

  1. Indicators of health status and antioxidant potential. The manuscript presents the results of research on vitamin C. Research methods are not presented.

Answer: The manuscript was completed with missing methods, please see line 167-196.

  1. Table 3. Why "0 day"?

Answer: Corrected to 'day 1', please see line 223.

  1. Figure 1. The figure does not clearly show the statistical error. Please change the picture.

Answer: All graphs in the manuscript were corrected according to the reviewer's recommendations.

  1. Table 4. Gizzard, g per, Specify - this is a muscular stomach?

Answer: Yes. This is a part of the bird's stomach that has thick muscles, where food is ground down. This term is commonly used in scientific papers.

  1. Table 4. Why is statistical processing of data not presented? Please correct the data.

Answer: Added: Data are given as mean ± SEM (n = 10). The commonly used method of presenting data in a table for the journal Antioxidants was used, please see line 232

Table 5. Why is statistical processing of data not presented? Please correct the data.

Answer: Added: Data are given as mean ± SEM (n = 10). The commonly used method of presenting data in a table for the journal Antioxidants was used, please see line 244

  1. Table 6. Why is statistical processing of data not presented? Please correct the data. In connection with what is the study of only cockerels (males) connected?

Answer: Added: Data are given as mean ± SEM (n = 10). The commonly used method of presenting data in a table for the journal Antioxidants was used, please see line 386.

  1. Table 7. The format of this table is not well chosen.

Answer: The table format has been improved, please see line 386.

  1. Discussion: It is necessary to clearly correspond to the purpose of the study. I did not receive an answer to the question of how the antioxidant potential is improved.

Answer: The missing passage has been completed, please see line 613-625

  1. Conclusions: It is necessary to significantly (expand) in accordance with the supplement purpose of the study.

Answer: The missing passage has been completed, please see line 627-637.

  1. References: It is desirable to transfer this quotation from the References section to the Materials and Methods section and indicate it in the text of the manuscript: 10. Aviagen Ross 308: Broiler Performance Objectives. Aviagen Inc., Huntsville, AL. 2019, 1–15

Answer: Corrected as recommended by the Reviewer, please see line 122-124.

Round 2

Reviewer 1 Report

Revisions made by authors are adequate.

Reviewer 2 Report

I recommend for publication the scientific article " Nano-sized selenium maintains performance, improves health status and antioxidant potential while not compromising ultrastructure of breast muscle and liver in chickens " - antioxidants-2296557